# Positive Associations between Body Mass Index and Hematological Parameters, Including RBCs, WBCs, and Platelet Counts, in Korean Children and Adolescents

**DOI:** 10.3390/children9010109

**Published:** 2022-01-14

**Authors:** Hwal Rim Jeong, Hae Sang Lee, Young Suk Shim, Jin Soon Hwang

**Affiliations:** 1Department of Pediatrics, School of Medicine, Soonchunhyang University, Cheonan Hospital, Cheonan 31151, Korea; hrjeong@schmc.ac.kr; 2Department of Pediatrics, School of Medicine, Ajou University Hospital, Suwon 16499, Korea; seaon98@naver.com (H.S.L.); pedhwang@ajou.ac.kr (J.S.H.)

**Keywords:** obesity, children, WBC, RBC, platelets

## Abstract

We conducted this study to investigate the associations between hematological parameters and obesity in children and adolescents. The levels of hematological parameters (including white blood cells [WBCs], red blood cells [RBCs], hemoglobin [Hb], hematocrit [Hct], and platelets) of 7997 participants (4259 boys and 3738 girls) aged 10–18 years were recorded. The parameters were compared among participants with normal weight, overweight, and obesity. Significantly higher mean levels of WBCs (7.16 vs. 6.16 × 103/mm^3^, *p* < 0.001), RBCs (4.90 vs. 4.82 × 106/mm^3^, *p* < 0.001), Hb (14.07 vs. 13.99 g/dL, *p* < 0.05), Hct (42.31 vs. 41.91%, *p* < 0.001), and platelets (311.87 vs. 282.66 × 103/mm^3^, *p* < 0.001) were found in the obese than normal weight group, respectively, after adjusting for body mass index (BMI) and sex. BMI SDS had significant positive associations with the levels of WBCs (β = 0.275, *p* < 0.001), RBCs (β = 0.028, *p* < 0.001), Hb (β = 0.034, *p* < 0.001), Hct (β = 0.152, *p* < 0.001), and platelets (β = 8.372, *p* < 0.001) after adjusting for age, sex, and socioeconomic factors in a multiple linear regression analysis. A higher BMI was associated with elevated WBC, RBC, Hb, Hct, and platelet counts in children and adolescents. Because higher levels of hematological parameters are potential risk factors for obesity-related diseases, hematological parameters should be evaluated in obese children and adolescents.

## 1. Introduction

Obesity is characterized by excessive body fat mass or weight. The prevalence of obesity has increased significantly worldwide, including among children [1]. Obesity is associated with chronic inflammation, which contributes to atherosclerosis and metabolic syndrome (MS) [2,3,4]. MS refers to a constellation of abnormalities, including central obesity, glucose intolerance, hypertriglyceridemia, decreased high-density lipoprotein (HDL) cholesterol levels, and high blood pressure [5]. MS is a precursor of cardiovascular disease and type 2 diabetes mellitus (T2DM) [6]. Childhood obesity leads to adult obesity and MS [7,8] and is associated with increased risks of cardiovascular disease and mortality [9]. Therefore, prevention and early treatment of childhood obesity and related comorbidities are important challenges for public health care systems.

Obesity-induced inflammation leads to insulin resistance and T2DM [3,10], which contribute to MS [11]. The levels of inflammatory markers, including C-reactive protein, ferritin, and cytokines, are elevated in obesity [11,12]. The total white blood cell (WBC) count, an inflammatory marker, is elevated in obesity and MS [13,14,15]. In addition, increased platelet count and activation occur as part of chronic inflammation in obesity [16,17]. Furthermore, obesity is a risk factor for high blood pressure [18], which is associated with elevated red blood cell (RBC) parameters, such as hemoglobin (Hb) and hematocrit (Hct) levels [19,20]. These findings suggest that changes in hematological parameters are accompanied by increased body mass and chronic inflammation in obesity.

Previous studies of the hematological changes in obesity have focused mainly on adults, and population studies of children are limited. The purpose of this study was to investigate the associations between body mass index (BMI) and hematologic parameters in children and adolescents.

## 2. Materials and Methods

We analyzed the data from the 2007–2018 Korea National Health and Nutrition Examination Survey (KNHANES), a cross-sectional and nationally representative survey with a multistage, stratified probability sampling design. The KNHANES survey has been conducted by the Division of Chronic Disease Surveillance, Korean Centers for Disease Control and Prevention, in 3-year cycles since 1998 to assess the health and nutritional statuses of the noninstitutionalized civilian population of Korea [21]. To enhance the statistical power of the analyses, we combined the data from the fourth (2007–2009), fifth (2010–2012), and sixth (2013–2015) survey, as well as the first and second years of the seventh survey (2017–2018). The details of the study design have been reported previously [22].

KNHANES enrolled 97,622 individuals between 2007 and 2017. Of these, our preliminary analyses included 10,734 (5670 boys, 5064 girls) aged 10–18 years (Figure 1). We attempted to include a large number of representative Korean children and adolescents. After obtaining informed consent, the participants and their parents were interviewed at their homes, and the participants underwent several examinations, including blood sampling, an oral examination, and pulmonary function tests. Different blood tests were conducted in each survey. The levels of glucose, HbA1c, total cholesterol, triglyceride, HDL cholesterol, LDL cholesterol (measured directly), HBs Ag, AST, ALT, HCV titer, BUN, and creatinine were measured annually. In the seventh survey from 2017 to 2018, the levels of hsCRP, uric acid, folate, vitamin A, and vitamin E were measured. In addition, the levels of urinary pH, nitrite, specific gravity, protein, glucose, ketone, cotinine, sodium, potassium, and heavy metals (including cadmium and mercury) were evaluated every year. The hematological parameters recorded were the WBC, RBC, Hb, Hct, and platelet count. Participants with incomplete physical examination records, including incomplete anthropometric measurements and laboratory test results (such as lipid profiles) or triglyceride levels ≥400 mg/dL, were excluded (*n* = 17). The KNHANES database is publicly available (http://knhabes.cdc.go.kr, accessed on 4 August 2021). The 2007–2018 KNHANES study protocols were approved by the Institutional Review Board of the Korean Centers for Disease Control and Prevention. Informed consent was provided by the KNHANES participants.

Anthropometric assessments, including height, weight, waist circumference (WC), and systolic and diastolic BP (SBP and DBP, respectively), were performed by an expert and recorded annually. Height was measured to the nearest 0.1 cm using the standard method on a flat floor without shoes or bulky clothing. The participants stood with their back toward the measuring rod, face forward, feet placed together, knees straight, heels touching the heel plate or wall, and shoulders, buttocks, and head inside the stadiometer (seca, Hamburg, Germany). Body weight was measured to the nearest 0.1 kg while wearing light clothing using an electronic scale (G-tech, Seoul, Korea). WC was measured to the nearest 0.1 cm at the level between the lower rib margin and iliac crest using a calibrated measuring tape (seca). SBP and DBP were measured three times at an interval of 2 min to the nearest 1 mmHg from the right upper arm using a calibrated sphygmomanometer. The mean values of the final two SBP and DBP measurements were used for analysis. The standard deviation scores (SDSs) for height, weight, WC, and BMI were calculated using age- and sex-specific least mean square parameters based on the 2017 growth reference values for Korean children and adolescents, developed by the Korean Pediatric Society and the Korea Centers for Disease Control and Prevention [23]. Based on the BMI, participants were categorized into normal weight (NW; BMI < 85th percentile), overweight (OW; BMI 85–95th percentile), and obese (OB; BMI ≥ 95th percentile) groups.

Lifestyle-related behaviors, such as alcohol consumption, smoking, household income, and residence area, were assessed using a questionnaire. Information about alcohol consumption (drinkers vs. nondrinkers) and smoking status (smokers vs. nonsmokers) was collected from participants aged ≥12 years using a self-administered questionnaire. Participants were also categorized based on physical activity (yes or no); those who performed intense physical activity for ≥20 min/day and ≥3 days/week or moderate physical activity for ≥30 min/day and ≥5 days/week, or who walked for ≥30 min/day and ≥5 days/week were included in the “yes” group.

Questionnaires related to household income, residence area (urban vs. rural), and previous diagnoses (T2DM, hypertension, and dyslipidemia) were administered by trained interviewers.

## 3. Statistical Analyses

R software (ver. 3.5.1; R Foundation for Statistical Computing, Vienna, Austria) was used for the statistical analysis. Continuous variables are expressed as means and standard deviations (SDs). Categorical variables are presented as numbers and percentages (%). Differences were analyzed using analysis of variance (ANOVA) for continuous variables and chi-square test for categorical variables. The adjusted means and standard errors (SEs) of the hematological indices (i.e., WBC, RBC, platelet, Hb, and Hct levels) were compared among the BMI groups using analysis of covariance (ANCOVA) after adjustment for possible confounders. In ANCOVA model 1, the adjusted means and SEs of hematological indices were estimated after controlling for age and sex. In ANCOVA models 2 and 3, the adjusted means and SEs of hematological indices were estimated for boys and girls after controlling for age. The pairwise differences among the BMI groups were tested for significance using *post-hoc* tests with Bonferroni correction in each ANCOVA model. To evaluate the correlations between hematological parameters and BMI SDS, Pearson’s correlation coefficient analysis with age and sex adjustments was performed. To evaluate the relationships between BMI SDS and hematological indices, multiple linear regression analyses were performed after adjusting for age, sex, alcohol consumption, smoking, physical activity, rural residence, household income, diagnosis of T2DM, hypertension, and dyslipidemia. The corresponding standardized regression coefficient (β) and SE were estimated. *p*-values < 0.05 were considered to indicate statistical significance.

## 4. Results

The final analyses included 7997 individuals: 4259 (53.26%) boys and 3738 girls. Participants were divided into the NW (6421, 80.29%), OW (782, 9.78%), and OB (794, 9.93%) groups. Among the 4259 boys, 3350 (78.66%), 443 (10.40%), and 466 (10.94%) were included in the NW, OW, and OB groups. Among the 3738 girls, 3071 (82.16%), 339 (9.07%), and 328 (8.77%) were included in the NW, OW, and OB groups.

### 4.1. Clinical Characteristics According to BMI

Table 1 shows the clinical characteristics of the study participants according to BMI. Mean SDSs for height, weight, BMI, and WC and mean SBP and DBP were significantly different among the subgroups (*p* < 0.001 for all). OB individuals had higher mean levels of hematological parameters and serum concentrations of glucose, total cholesterol, triglycerides, low-density lipoprotein cholesterol (LDL-C), and a lower serum concentration of HDL-C compared with non-OB individuals. The rate of alcohol use was higher in the OB group than in the other groups. The clinical characteristics of the study participants according to weight and sex are presented in Appendix A.

The mean hematological parameters according to sex and BMI are shown in Figure 2. The mean WBC count was significantly elevated in boys and girls in all groups (Table 2). In both sexes, the OB group showed significantly higher mean RBC and Hct levels compared with the NW group. The mean Hb level was significantly higher in the OB than NW group in boys but not girls. Mean platelet counts in the OW and OB groups were significantly higher than that in the NW group for both sexes.

### 4.2. Hematological Parameters Adjusted by Sex and Obesity

The adjusted mean levels of hematological parameters were significantly different among the BMI groups (Table 3). OB participants had higher adjusted mean levels of WBCs (7.16 vs. 6.16 × 10^3^/mm^3^), RBCs (4.90 vs. 4.82 × 10^6^/mm^3^), Hct (42.31% vs. 41.91%), and platelets (311.87 vs. 282.66 × 10^3^/mm^3^) (*p* < 0.001 for all) compared with NW participants. Similarly, OB girls had higher adjusted mean levels of WBCs, RBCs, Hct, and platelets (*p* < 0.001 for all) compared with NW girls. Although the Hb level was significantly different among the three groups, the post-hoc tests did not reveal a significant difference in all participants and girls. OB boys had higher adjusted mean levels of WBCs, RBCs, Hb, Hct, and platelets than those of NW boys (*p* < 0.001 for all).

### 4.3. Correlations between BMI SDS and Hematological Indices

The unadjusted and adjusted correlations between BMI SDS and hematological parameters are presented in Table 4. BMI SDS was positively correlated with the levels of WBCs (*r* = 0.222), RBCs (*r* = 0.109), Hb (*r* = 0.042), Hct (*r* = 0.067), and platelets (*r* = 0.180) after adjusting for age and sex (*p* < 0.001 for all). BMI SDS was not correlated with the Hb level in girls in the unadjusted model but was significantly correlated after adjusting for age. After adjusting for age, BMI SDS was positively correlated with hematological indices in both sexes. Pearson’s coefficients for the correlations of BMI SDS with RBC, Hb, and Hct levels in girls were less than half of those in boys.

### 4.4. Multiple Linear Regression Analyses of the BMI Groups with Hematological Parameters

Table 5 shows that BMI SDS had significant positive associations with hematological parameters among all participants after adjusting for age, sex, alcohol consumption, smoking, physical activity, rural residence, household income, T2DM, hypertension, and dyslipidemia. BMI SDS was significantly associated with WBC (β = 0.275), RBC (β = 0.028), Hb (β = 0.034), Hct (β = 0.152), and platelet (β = 8.372) levels (*p* < 0.001 for all). In boys, BMI SDS was significantly associated with WBC (β = 0.279), RBC (β = 0.043), Hb (β = 0.073), Hct (β = 0.267), and platelet (β = 7.658) levels (*p* < 0.001 for all). In girls, BMI SDS was associated with WBC (β = 0.270, *p* < 0.001), RBC (β = 0.019, *p* < 0.001), Hb (β = 0.027, *p* = 0.038), Hct (β = 0.118, *p* < 0.001), and platelet (β = 8.715, *p* < 0.001) levels. The regression coefficients between BMI SDS and RBC, Hb, and Hct levels were less than half of those in boys.

## 5. Discussion

The current study investigated the relationships between BMI and hematological parameters in Korean children and adolescents. OB children had higher blood pressure, glucose, LDL-C, WBC, RBC, and platelet levels. In particular, BMI SDS had an independent positive association with hematological parameters after adjusting for cofounding variables. To the best of our knowledge, this is the largest population-based study of the relationships between BMI and hematological parameters in Korean children and adolescents.

Chronic inflammation around adipocytes plays an important role in obesity-related diseases [3]. Because the WBC count is increased in inflammation, it is likely that the WBC count is elevated in OB patients. A plausible explanation for the increased WBC count in OB individuals is that adipose tissue produces IL-6, a proinflammatory cytokine involved in bone marrow granulopoiesis and WBC proliferation and differentiation [24,25]. In our study, the WBC count in children increased by 0.275 × 10^3^/mm^3^ for every 1-point increase in BMI SDS. The elevated WBC count observed in the present study is consistent with previous studies [26,27]. The elevated WBC count was associated with carotid atherosclerosis and impaired glucose tolerance in previous studies [28]. Tong et al. reported that patients with high WBC counts had adverse metabolic profiles, even when the WBC levels were high but within the normal range. The elevated WBC count is associated with macrovascular and microvascular complications of T2DM [29]. Furthermore, Veronelli et al. observed a significant decrease in the WBC count after bariatric surgery, suggesting the usefulness of weight loss in reducing the WBC count in morbidly OB individuals [27].

Our findings are consistent with those of previous studies that found higher mean WBC counts and MS prevalence with increasing BMI in boys and girls [13,30]. In Colombian children, the WBC count was associated with truncal adiposity [12]. An increased WBC count was associated with early derangement of glucose metabolism and preclinical signs of liver, vascular, and cardiac damage in Italian children [31]. Park et al. reported that a higher WBC count was positively associated with an increased risk of insulin resistance in Korean children and adolescents [32]. Lee et al. suggested that an elevated WBC count is a surrogate marker of MS in Korean children and adolescents [13]. Because it is inexpensive, the WBC count has been suggested as an effective tool for identifying OB children at risk of complications [31,32].

In this study, red cell indices, including RBC count Hb and Hct levels, were positively associated with BMI SDS, which was consistent with previous studies. Mărginean et al. reported a significantly higher RBC count but no significant difference in the Hb level in OB children than controls [33]. In addition, several studies suggested that the RBC count has a significant correlation with MS [34,35], and the Hb level is significantly associated with high blood pressure [36]. The mechanism underlying the increased RBC cell indices in obesity is not known. However, iron deficiency anemia occurs more frequently in OB and OW individuals compared with NW individuals [37], which is likely related to an obesity-induced chronic inflammatory state and effects of hepcidin. Bekri et al. reported that hepcidin, a proinflammatory adipokine, reduces iron bioavailability by controlling the ferroportin-1 exporter, resulting in severe iron deficiency anemia in OB individuals [38]. Therefore, obesity-induced chronic inflammation may influence the serum iron level. Ausk et al. reported that the serum Hb concentration was not significantly different between OB and NW individuals [39]; however, a higher serum ferritin level was associated with higher BMI and lower serum iron and transferrin saturation levels. The study researchers concluded that OW and OB individuals were no more likely to be anemic than were NW individuals. In our study, BMI SDS was independently associated with RBC indices after adjusting for multiple variables. However, the serum iron level, nutritional habits, and anemia prevalence were not compared among the BMI subgroups. Further studies conducted in different ethnic groups are needed to validate our results.

The changes in RBC indices differ between males and females. Kim et al. reported that the RBC count was significantly increased in men, but not women, with MS [40]. In the present study, all RBC indices increased with increasing BMI SDS, but the degree of increase differed between boys and girls. The RBC count increased by 0.043 × 10^6^/mm^3^ in boys and 0.019 × 10^6^/mm^3^ in girls with every 1-point increase in BMI SDS. The RBC count was over 2-fold higher in boys than girls. Similar patterns were observed for Hb and Hct levels. Obesity is a chronic hypoxia state that causes adipose tissue dysfunction, inflammation, and insulin resistance [41]. Chronic hypoxia may lead to increased production of RBCs and WBCs. In addition, the level of ferritin, an inflammatory marker, is increased in obesity [11], which may be involved in the changes in RBC indices. Although adolescent girls typically have more body fat and less muscle than boys, they periodically lose blood and iron due to menstruation. Consequently, the increase in RBC count with increasing BMI SDS may be relatively less for girls than boys.

Thrombocytosis suggests inflammation, and platelet activation leads to accelerated atherothrombosis [17]. Lim et al. reported that an elevated platelet count was associated with an increased prevalence and risk of MS in children and adolescents [42]. In our study, for every 1-point increase in BMI SDS, the platelet count increased by 8.372 × 10^3^/mm^3^ for all participants, 8.715 × 10^3^/mm^3^ for girls, and 7.658 × 10^3^/mm^3^ for boys. The increase in the platelet count also differed by sex. Dorit et al. reported that OB females had a significantly higher platelet count compared with NW females [16]; however, the platelet count was not elevated in males. In addition, Charles et al. reported positive associations of obesity with WBC and platelet counts among female officers, but not male officers [43]. In our study, BMI SDS was positively associated with hematological parameters; however, the changes in hematological parameters (especially RBC and platelet counts) showed differences between the sexes (i.e., sexual dimorphism). The adjusted mean RBC indices and changes thereof with BMI SDS were higher in boys than girls for all subgroups. In contrast, the mean platelet count and increase in platelet count with increasing BMI SDS were higher in girls than boys in all subgroups. The mechanisms underlying the sex differences in blood cell composition and their effects on the risk of obesity-related complications are not well understood. It is possible that these findings result from differences in body fat composition between males and females. Nuttall reported that BMI may not be an accurate marker of obesity because men tend to accumulate fat in the abdominal area, whereas women tend to accumulate it in the peripelvic area and thighs [44]. Before puberty, boys and girls have similar patterns of body fat deposition; however, during and after puberty, girls tend to accumulate a large quantity of fat, whereas boys accumulate a large quantity of lean mass (bone and muscle) but not fat mass. These changes lead to an increased BMI in both sexes. The hematological parameters of boys and girls are different between those tested before and after puberty. In the present study, information related to body fat mass and pubertal status was not available. Therefore, further studies are needed to determine the sex differences in hematological parameters before and after puberty.

This study had several limitations. First, because this was a cross-sectional study, we could not identify causal relationships between obesity and hematological parameters. Second, our study included children and adolescents aged 10–18 years; therefore, no data are available for children aged < 10 years. Third, because data regarding body composition and pubertal status were not available, the associations of body fat mass with hematological parameters were not analyzed. In addition, the effects of puberty on hematological changes and body composition were not evaluated. Importantly, BMI cannot differentiate between body lean and fat mass and poorly represents the body fat percentage and location [44]. Finally, we could not determine the mechanisms underlying the relationships between BMI SDS and hematological indices. Our results may not be applicable to specific populations, such as those with severe obesity and malnutrition. Several studies showed that OW and OB children have an increased risk of iron deficiency anemia [45,46]. Despite the limitations, this study showed that BMI SDS was independently associated with hematological parameters in a relatively large number of children and adolescents. Our findings provide insight into the hematologic changes in OB children and adolescents.

In conclusion, this nationally representative population-based study showed that higher BMI was independently associated with high levels of hematological parameters, including WBC, RBC, Hb, Hct, and platelet levels, in children and adolescents. Our results suggest that obesity is related to hematological changes, which may predispose to obesity-related diseases. When interpreting the complete blood count in children and adolescents, it is important to consider the effects of BMI and sex.

## Figures and Tables

**Figure 1 children-09-00109-f001:**
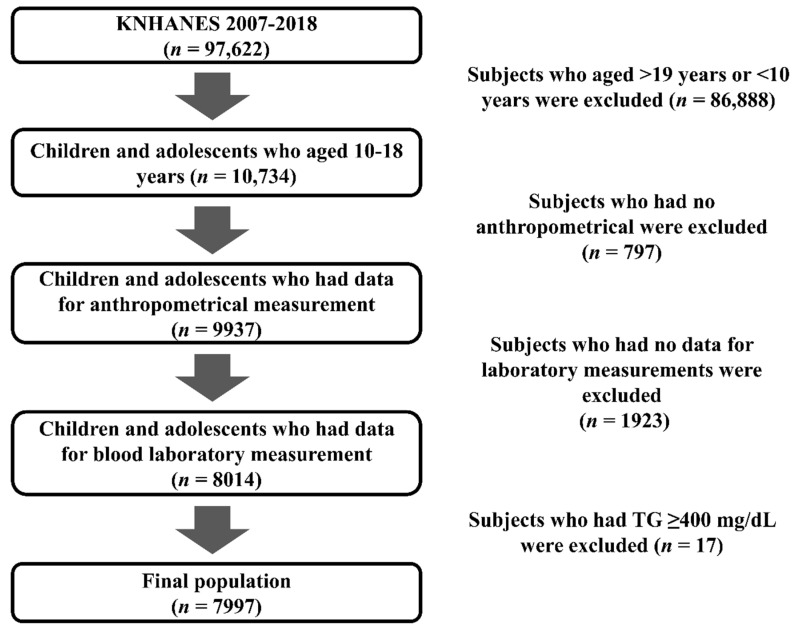
Flow chart of study population.

**Figure 2 children-09-00109-f002:**
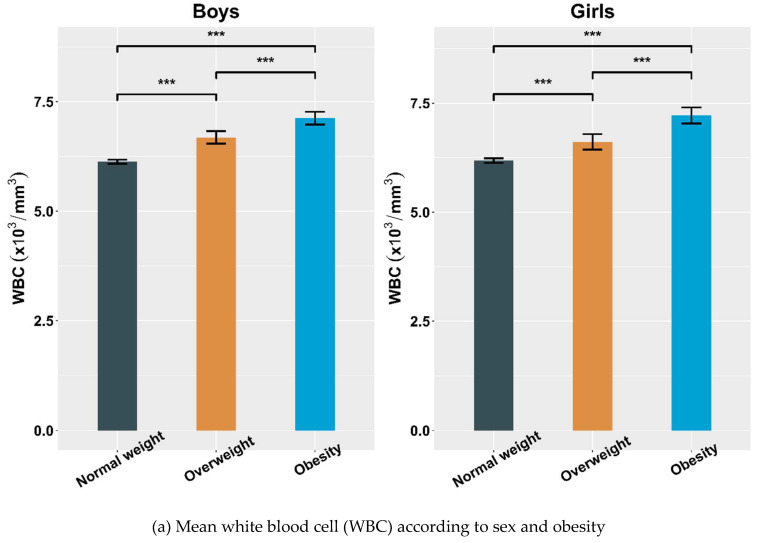
Mean white blood cell (WBC), red blood cell (RBC), hemoglobin, hematocrit, and platelet levels according to sex and obesity. (**a**), Mean white blood cell (WBC) according to sex and obesity; (**b**), Mean red blood cell (RBC) according to sex and obesity; (**c**), Mean hemoglobin levels according to sex and obesity; (**d**), Mean hematocrit levels according to sex and obesity; (**e**), Mean platelet levels according to sex and obesity; *, *p* < 0.05; **, *p* < 0.005; ***, *p* < 0.001; NS., *p* > 0.05.

**Table 1 children-09-00109-t001:** Clinical characteristics of the study participants by body weight (*n* = 7997).

	Normal Weight	Overweight	Obesity	*p*
Total, *n* (%)	6421 (80.29%)	782 (9.78%)	794 (9.93%)	
Boys, *n* (%)	3350 (52.17%)	443(56.65%)	466 (58.69%)	<0.001
Age (year)	14.33 ± 2.51	14.24 ± 2.51	14.78 ± 2.54	<0.001
Height SDS	0.17 ± 1.03	0.47 ± 1.04	0.53 ± 1.11	<0.001
Weight SDS	−0.31 ± 0.89	1.29 ± 0.49	2.20 ± 0.74	<0.001
BMI SDS (kg/m^2^)	−0.59 ± 0.88	0.86 ± 0.45	1.64 ± 0.75	<0.001
WC SDS	−0.49 ± 0.85	1.32 ± 0.18	2.40 ± 0.67	<0.001
SBP (mmHg)	105.65 ± 9.77	109.94 ± 10.28	113.68 ± 10.95	<0.001
DBP (mmHg)	65.59 ± 8.91	66.88 ± 8.88	69.07 ± 9.19	<0.001
WBC (×10^3^/mm^3^)	6.16 ± 1.49	6.65 ± 1.58	7.16 ± 1.65	<0.001
RBC (×10^6^/mm^3^)	4.81 ± 0.39	4.89 ± 0.40	4.93 ± 0.41	<0.001
Hemoglobin (g/dL)	13.97 ± 1.23	14.07 ± 1.22	14.19 ± 1.30	<0.001
Hematocrit (%)	41.86 ± 3.40	42.28 ± 3.42	42.65 ± 3.53	<0.001
Platelets (×10^3^/mm^3^)	282.96 ± 58.58	296.81 ± 59.92	309.03 ± 63.53	<0.001
Glucose (mg/dL)	90.01 ± 7.39	91.83 ± 11.05	92.45 ± 12.16	<0.001
T-C (mg/dL)	158.25 ± 26.27	163.00 ± 28.46	169.35 ± 29.50	<0.001
HDL-C (mg/dL)	52.16 ± 9.94	47.68 ± 8.63	44.99 ± 8.37	<0.001
TG (mg/dL)	79.29 ± 41.79	99.34 ± 54.29	112.75 ± 58.76	<0.001
LDL-C (mg/dL)	90.24 ± 22.42	95.46 ± 25.07	101.81 ± 25.63	<0.001
Alcohol consumption	1564 (24.36%)	188 (24.04%)	237 (29.85%)	0.003
Smoker	718 (11.18%)	88 (11.25%)	106 (13.35%)	0.191
Household income ≤ 1st quartile	687 (10.70%)	82 (10.49%)	95 (11.96%)	0.531
Rural residence	985 (15.34%)	113 (14.45%)	125 (15.74%)	0.757
Physical activity	2368 (36.88%)	279 (35.68%)	316 (39.80%)	0.194
Hypertension	1 (0.02%)	1 (0.13%)	1 (0.13%)	0.124
T2DM	0 (0%)	0 (0%)	0 (0%)	>0.999
Dyslipidemia	0 (0%)	0 (0%)	0 (0%)	>0.999

SDS, standard deviation score; BMI, body mass index; WC, waist circumference; SBP, systolic blood pressure; DBP, diastolic blood pressure; WBC, white blood cell; RBC, red blood cell; T-C, total cholesterol; HDL-C, high-density lipoprotein cholesterol; TG, triglycerides; LDL-C, low-density lipoprotein cholesterol; T2DM, type 2 diabetes mellitus.

**Table 2 children-09-00109-t002:** Mean white blood cell (WBC), red blood cell (RBC), hemoglobin, hematocrit, and platelet levels according to sex and obesity.

	Boys			Girls		
	NW	OW	OB	NW	OW	OB
	*n* = 3350	*n* = 443	*n* = 466	*n* = 3071	*n* = 339	*n* = 328
WBC (×10^3^/mm^3^)	6.13 ± 1.45	6.68 ± 1.53 ^a^	7.13 ± 1.60 ^b,c^	6.18 ± 1.53	6.61 ± 1.64 ^a^	7.22 ± 1.71 ^b,c^
RBC (×10^6^/mm^3^)	5.02 ± 0.34	5.10 ± 0.34 ^a^	5.13 ± 0.33 ^c^	4.59 ± 0.31	4.62 ± 0.30	4.63 ± 0.31 ^c^
Hemoglobin (g/dL)	14.60 ± 1.11	14.64 ± 1.15	14.79 ± 1.20 ^c^	13.28 ± 0.96	13.33 ± 0.85	13.34 ± 0.92
Hematocrit (%)	43.43 ± 3.30	43.73 ± 3.38	44.18 ± 3.34 ^c^	40.15 ± 2.60	40.38 ± 2.42	40.49 ± 2.50 ^c^
Platelet (×10^3^/mm^3^)	278.59 ± 58.48	295.33 ± 61.62 ^a^	302.91 ± 61.66 ^c^	287.73 ± 58.32	298.74 ± 57.65 ^a^	317.73 ± 65.21 ^b,c^

Data are presented as the mean ± standard deviation (SD). NW, underweight and normal weight; OW, overweight; OB, obesity; WBC, white blood cell; RBC, red blood cell. Bonferroni’s *post-hoc* test after adjustment for age among girls.^a^: *p* < 0.05, NW group versus OW group after Bonferroni’s post-hoc test. ^b^: *p* < 0.05, OW group versus OB group after Bonferroni’s post-hoc test. ^c^: *p* < 0.05, NW group versus OB group after Bonferroni’s post-hoc test.

**Table 3 children-09-00109-t003:** Adjusted mean levels of white blood cells, red blood cells, hemoglobin, hematocrit, and platelets by sex and BMI.

All Participants ^1^ (*n* = 7997)	Normal Weight	Overweight	Obesity	*p* for Trend
	*n* = 6421	*n* = 782	*n* = 794	
WBC (×10^3^/mm^3^)	6.16 ± 0.02	6.66 ± 0.05 ^a^	7.16 ± 0.05 ^b,c^	<0.001
RBC (×10^6^/mm^3^)	4.82 ± 0.01	4.88 ± 0.01 ^a^	4.90 ± 0.01 ^c^	<0.001
Hemoglobin (g/dL)	13.99 ± 0.01	14.04 ± 0.04	14.07 ± 0.04	0.016
Hematocrit (%)	41.91 ± 0.04	42.21 ± 0.10 ^a^	42.31 ± 0.10 ^c^	<0.001
Platelets (×10^3^/mm^3^)	282.66 ± 0.72	296.38 ± 2.05 ^a^	311.87 ± 2.04 ^b,c^	<0.001
**Boys ^2^**				
(*n* = 4259)	*n* = 3350	*n* = 443	*n* = 466	
WBC (×10^3^/mm^3^)	6.13 ± 0.03	6.69 ± 0.07 ^a^	7.12 ± 0.07 ^b,c^	<0.001
RBC (×10^6^/mm^3^)	5.02 ± 0.01	5.11 ± 0.01 ^a^	5.12 ± 0.01 ^c^	<0.001
Hemoglobin (g/dL)	14.60 ± 0.01	14.72 ± 0.04 ^a^	14.72 ± 0.04 ^c^	<0.001
Hematocrit (%)	43.43 ± 0.04	43.96 ± 0.12 ^a^	44.00 ± 0.12 ^c^	<0.001
Platelets (×10^3^/mm^3^)	278.64 ± 0.97	293.21 ± 2.66 ^a^	304.58 ± 2.59 ^b,c^	<0.001
**Girls ^3^**				
(*n* = 3738)	*n* = 3071	*n* = 339	*n* = 328	
WBC (×10^3^/mm^3^)	6.18 ± 0.03	6.61 ± 0.08 ^a^	7.20 ± 0.09 ^b,c^	<0.001
RBC (×10^6^/mm^3^)	4.59 ± 0.01	4.62 ± 0.02	4.66 ± 0.02 ^c^	<0.001
Hemoglobin (g/dL)	13.27 ± 0.02	13.34 ± 0.05	13.39 ± 0.05	0.015
Hematocrit (%)	40.14 ± 0.05	40.39 ± 0.14	40.59 ± 0.14 ^c^	<0.001
Platelets (×10^3^/mm^3^)	287.47 ± 1.05	298.90 ± 3.17 ^a^	319.95 ± 3.23 ^b,c^	<0.001

Data are presented as the mean ± standard error (SE). WBC, white blood cell; RBC, red blood cell. Model 1: Comparisons with BMI using analysis of covariance with Bonferroni’s *post-hoc* test after adjustment for age and sex among all participants. Model 2: Comparisons with BMI using analysis of covariance with Bonferroni’s *post-hoc* test after adjustment for age among boys. Model 3: Comparisons with BMI using analysis of covariance with Bonferroni’s *post-hoc* test after adjustment for age among girls. ^a^: *p* < 0.05 between normal weight vs. overweight group after Bonferroni’s *post-hoc* test. ^b^: *p* < 0.05 between overweight vs. obesity group after Bonferroni’s *post-hoc* test. ^c^: *p* < 0.05 between normal weight vs. obesity group after Bonferroni’s *post-hoc* test.

**Table 4 children-09-00109-t004:** Unadjusted and adjusted correlations between body mass index (BMI) standard deviation score (SDS) and white blood cell (WBC), red blood cell (RBC), hemoglobin, hematocrit, and platelet levels.

All Participants (*n* = 7997)	*r* ^1^	*p*	*r* ^2^	*p*
WBC (×10^3^/mm^3^)	0.223	<0.001	0.222	<0.001
RBC (×10^6^/mm^3^)	0.096	<0.001	0.109	<0.001
Hemoglobin (g/dL)	0.044	<0.001	0.042	<0.001
Hematocrit (%)	0.067	<0.001	0.067	<0.001
Platelets (×10^3^/mm^3^)	0.169	<0.001	0.180	<0.001
**Boys** (*n* = 4259)	*r* ^3^	*p*	*r* ^4^	*p*
WBC (×10^3^/mm^3^)	0.239	<0.001	0.240	<0.001
RBC (×10^6^/mm^3^)	0.157	<0.001	0.180	<0.001
Hemoglobin (g/dL)	0.070	<0.001	0.111	<0.001
Hematocrit (%)	0.090	<0.001	0.137	<0.001
Platelets (×10^3^/mm^3^)	0.175	<0.001	0.177	<0.001
**Girls** (*n* = 3738)	*r* ^5^	*p*	*r* ^6^	*p*
WBC (×10^3^/mm^3^)	0.206	<0.001	0.202	<0.001
RBC (×10^6^/mm^3^)	0.047	0.004	0.078	<0.001
Hemoglobin (g/dL)	0.014	0.383	0.033	0.043
Hematocrit (%)	0.042	0.009	0.054	<0.001
Platelets (×10^3^/mm^3^)	0.164	<0.001	0.176	<0.001

WBC, white blood cell; RBC, red blood cell. ^1^: Pearson’s correlation analyses were conducted between BMI SDS and WBC, RBC, hemoglobin, hematocrit, and platelet levels with no adjustments. ^2^: Pearson’s correlation analyses were conducted between BMI SDS and WBC, RBC, hemoglobin, hematocrit, and platelet levels after adjustments for age and sex for all participants. ^3^: Pearson’s correlation analyses were conducted between BMI SDS and WBC, RBC, hemoglobin, hematocrit, and platelet levels with no adjustments in boys. ^4^: Pearson’s correlation analyses were conducted between BMI SDS and WBC, RBC, hemoglobin, hematocrit, and platelet levels after adjustment for age in boys. ^5^: Pearson’s correlation analyses were conducted between BMI SDS and WBC, RBC, hemoglobin, hematocrit, and platelet levels with no adjustments. ^6^: Pearson’s correlation analyses were conducted between BMI SDS and WBC, RBC, hemoglobin, hematocrit, and platelet levels after adjustment for age in girls.

**Table 5 children-09-00109-t005:** Multiple regression analysis of the associations of body mass index (BMI) standard deviation score (SDS) with white blood cell (WBC), red blood cell (RBC), hemoglobin, hematocrit, and platelet levels.

All Participants ^1^ (*n* = 7997)	Β	SE	*p*
WBC (×10^3^/mm^3^)	0.275	0.013	<0.001
RBC (×10^6^/mm^3^)	0.028	0.003	<0.001
Hemoglobin (g/dL)	0.034	0.009	<0.001
Hematocrit (%)	0.152	0.025	<0.001
Platelet (×10^3^/mm^3^)	8.372	0.510	<0.001
**Boys** ^2^ (*n* = 4259)			
WBC (×10^3^/mm^3^)	0.279	0.017	<0.001
RBC (×10^6^/mm^3^)	0.043	0.004	<0.001
Hemoglobin (g/dL)	0.073	0.010	<0.001
Hematocrit (%)	0.267	0.030	<0.001
Platelet (×10^3^/mm^3^)	7.658	0.657	<0.001
**Girls** ^3^ (*n* = 3738)			
WBC (×10^3^/mm^3^)	0.270	0.021	<0.001
RBC (×10^6^/mm^3^)	0.019	0.004	<0.001
Hemoglobin (g/dL)	0.027	0.013	0.038
Hematocrit (%)	0.118	0.035	<0.001
Platelet (×10^3^/mm^3^)	8.715	0.798	<0.001

WBC, white blood cell; RBC, red blood cell. ^1^: Multiple linear regression analysis was conducted between BMI SDS (independent variable) and WBC, RBC, hemoglobin, hematocrit, and platelet levels (dependent variables) after adjustments for age, sex, alcohol consumption, smoking, physical activity, rural residence, household income, type 2 diabetes mellitus (T2DM), hypertension, and dyslipidemia for all participants. ^2^: Multiple linear regression analysis was conducted between BMI SDS (independent variable) and WBC, RBC, hemoglobin, hematocrit, and platelet levels (dependent variables) after adjustments for age, sex, alcohol consumption, smoking, physical activity, rural residence, household income, T2DM, hypertension, and dyslipidemia among boys. ^3^: Multiple linear regression analysis was conducted between BMI SDS (independent variable) and WBC, RBC, hemoglobin, hematocrit, and platelet levels (dependent variables) after adjustments for age, sex, alcohol consumption, smoking, physical activity, rural residence, household income, diagnosis of T2DM, hypertension, and dyslipidemia among girls.

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
