# Peer review of "Positive Associations between Body Mass Index and Hematological Parameters, Including RBCs, WBCs, and Platelet Counts, in Korean Children and Adolescents"

_children, 2022, doi:10.3390/children9010109_

Round 1

Reviewer 1 Report

Hwal Rim Jeong et al. created a publication based on many participants to investigate the associations between hematologic parameters and obesity in Korean children and adolescents. 

Major comments

The undoubted advantage of the work is a large number of examined boys and girls.

The morphology results in the three analyzed groups do not differ from the recommended range; only in the obese group are slightly higher, but still normal within the range. These conclusions which authors created cannot be drawn if the body composition is not examined. The percentage of fat significantly increase in overweight patients (normal range F - 16-25%, M - 15-20%; obese F 30%, M 25%) while the total amount of water (normal range 45-60%) significantly decreases (obesity less than 45%). When hydration is declining, it is usually essential to determine the intracellular and extracellular water ratio. The reduction of extracellular water usually increases the concentration of many parameters, which may justify the obtained results by the authors. Due to the decrease of water in favor of fat, it is worth assessing red cell parameters such as MCV, MCH, and MCHC because they would show anomalies in the intracellular space.

Another issue is the subclinical inflammation in obesity. Chronic inflammation causes normocytic or microcytic anemia. Various mechanisms are involved in this process, ranging from reduced iron absorption in GI tract to decreased red blood cells survival and many others causes. The authors do not mention this issue at all. The value of any parameter characterizing inflammation was not given in baseline characteristics of the patients, apart from WBCs. The authors mentioned in the introduction that obesity-related inflammation is essential.

The analyzed group of children and adults do not have comorbid disorders such as hypertension (presented only 0.02-0.13%), dyslipidemia, or diabetes. They are an excellent research group because obesity without comorbidities is very rare. Maybe it is because they are mainly young persons, not adults. 

It gives a false depiction of the assessment of morphology and its clinical implications in obese children and adults. The work should be supplemented in the results of total body composition especially including fat and water proportion. Changes in morphology should be analyzed concerning these measurements

Author Response

Major comments

The undoubted advantage of the work is a large number of examined boys and girls.

 à Thank you for your comment

The morphology results in the three analyzed groups do not differ from the recommended range; only in the obese group are slightly higher, but still normal within the range. These conclusions which authors created cannot be drawn if the body composition is not examined. The percentage of fat significantly increase in overweight patients (normal range F - 16-25%, M - 15-20%; obese F 30%, M 25%) while the total amount of water (normal range 45-60%) significantly decreases (obesity less than 45%). When hydration is declining, it is usually essential to determine the intracellular and extracellular water ratio. The reduction of extracellular water usually increases the concentration of many parameters, which may justify the obtained results by the authors. Due to the decrease of water in favor of fat, it is worth assessing red cell parameters such as MCV, MCH, and MCHC because they would show anomalies in the intracellular space.

 à Thank you for your helpful comments. We agree that the proportion of body water in obese individuals may influence red cell parameters, such as MCV, MCH, and MCHC. Unfortunately, the Korea National Health and Nutrition Examination Survey (KNHANES) did not evaluate red cell parameters. This is a limitation of our study, and future studies are needed to evaluate the red cell parameters.

Another issue is the subclinical inflammation in obesity. Chronic inflammation causes normocytic or microcytic anemia. Various mechanisms are involved in this process, ranging from reduced iron absorption in GI tract to decreased red blood cells survival and many others causes. The authors do not mention this issue at all. The value of any parameter characterizing inflammation was not given in baseline characteristics of the patients, apart from WBCs. The authors mentioned in the introduction that obesity-related inflammation is essential.

 à Thank you for your useful comments. Anemia is present in some obese patients. Several studies have reported that obesity may disrupt iron homeostasis, resulting in iron deficiency anemia. However, some studies have reported that overweight and obese individuals are no more likely than normal weight individuals to have anemia (Obesity (Silver Spring). 2008 Oct;16(10):2356-61). We have discussed these studies in the revised article (Discussion, lines 276–289). Our study results show that BMI SDS was independently associated with the RBC indices.

 Our study analyzed the results of the KNHANES. Participants of the KNHANES may differ from the general population because this survey did not include solely obese individuals. However, the results did not change even after adjustments for age, sex, alcohol consumption, smoking, physical activity, rural residence, household income, T2DM, hypertension, and dyslipidemia (Table 5). Therefore, the results will need to be verified in future studies that include other ethnicities and populations.

We have modified the text as follows:

 ‘Therefore, obesity-induced chronic inflammation may influence the serum iron level. Ausk et al. reported that the serum Hb concentration was not significantly different between OB and NW individuals; however, a higher serum ferritin level was associated with higher BMI and lower serum iron and transferrin saturation levels. The study researchers concluded that OW and OB individuals were no more likely to be anemic than were NW individuals. In our study, BMI SDS was independently associated with RBC indices after adjusting for multiple variables. However, the serum iron level, nutritional habits, and anemia prevalence were not compared among the BMI subgroups. Further studies conducted in different ethnic groups are needed to validate our results.’

Although it would have been useful to analyze the hsCRP level according to sex and BMI, the KNHANES dataset only provided hsCRP levels for the participants included between 2016 and 2019. Therefore, we did not include hsCRP as a covariate in the current analysis. However, the hsCRP levels of a limited number of participants according to sex and BMI are presented below:

Normal

(n=912)

Overweight

(n=132)

Obesity

(n=164)

P

Normal

(n=800)

Overweight

(n=99)

Obesity

(n=123)

P

hs-CRP

0.74 ± 1.60

1.21 ± 1.91

1.61 ± 1.86

<0.001

0.63 ± 1.32

0.87 ± 1.50

1.44 ± 2.12

<0.001

The analyzed group of children and adults do not have comorbid disorders such as hypertension (presented only 0.02-0.13%), dyslipidemia, or diabetes. They are an excellent research group because obesity without comorbidities is very rare. Maybe it is because they are mainly young persons, not adults. It gives a false depiction of the assessment of morphology and its clinical implications in obese children and adults. The work should be supplemented in the results of total body composition especially including fat and water proportion. Changes in morphology should be analyzed concerning these measurements

 à Thank you for your comment. We analyzed the results of the KNHANES, which is a large-scale population study conducted in randomly selected individuals. The survey included participants diagnosed with diabetes, dyslipidemia, and hypertension, as well as undiagnosed individuals.

Unfortunately, no data are available regarding body composition in the KNHANES population; therefore, the results could not be analyzed according to body composition.

We have added the following limitation to the Discussion.

‘ Third, because data regarding body composition and pubertal status were not available, the associations of body fat mass with hematological parameters were not analyzed. In addition, the effects of puberty on hematological changes and body composition were not evaluated. Importantly, BMI cannot differentiate between body lean and fat mass and poorly represents the body fat percentage and location’

Reviewer 2 Report

The study entitled “Positive association between body mass index and hematologic parameters, including RBC, WBC and platelet count, in Korean children and adolescent”, aims to investigate the associations between hematologic parameters and IMC in 7997 children and adolescents (4259 boys and 3738 girls) aged between 10 and 18 years old.

General Comments

This is a study that has its strong points in the large sample used and in the possibility of having been able to extract a wide variety of hematological variables. In this sense, the authors of the study are to be congratulated. On the other hand, although both the design and the methodology are also robust, there are a number of aspects that, in my opinion, would improve the information reported and are subject to major revision.

First aspect to review

The object of study revolves around the relationship of a series of variables with BMI, an index that, although standardized and commonly used, has multiple limitations given that it is based exclusively on quantitative aspects of body mass without being able to discern the qualitative aspects that determine it (FFM, FM, etc.). Given that the BMI is also limited when it comes to establishing comparisons between different age groups, it is necessary that, in the introduction section, the authors include a paragraph on the advantages, but especially the limitations of the BMI: Nuttall FQ. Body Mass Index: Obesity, BMI, and Health: A Critical Review. Nutr Today. 2015 May;50(3):117-128.

Second aspect to review.

The manuscript in general is not entirely clear in expressing, at the same level of analysis, the results recorded for boys and girls separately. This is because, although at certain points in the text and in the tables the results are separated by sex, it seems that the choice of when and when not to do so has been made erratically at the choice of the authors. An example is Table 1. It is not done (and it is required by this reviewer) to show the results (and the consequent statistical treatment) for both sexes, and also for the whole group. Example of the reconfiguration of Table 1:   

Boys (n=…)

Girls (n=…)

All (n=….)

Statistics (Boys vs Girls)

NW

OW

OB

NW

OW

OB

NW

OW

OB

NW

OW

OB

Age

Height

Weight

BMI

WC

In accordance with this, the entire manuscript should be revised to balance the expression of results, discussion and conclusions at the same level between both sexes. This is critical in this study because, as the authors themselves recognize, one of the aspects that most limit the consistency of their results and consequent interpretations, is the fact that no biological age variable has been recorded, a circumstance that is aggravated by the age range analyzed (10 - 18 years...), where the internal changes between boys and girls during puberty are significant and limit, to the point of being considered an error, to report them jointly....

Third aspect to review (or reflect on)

Although the authors recognize the limitations of the cross-sectional design, this could have been overcome by including a longitudinal follow-up subsample. That is, under a temporal inclusion criterion (e.g., including subjects analyzed for "X" years in a row), add their analysis. Undoubtedly, determining the "cause-effect" relationship of BMI behavior with the hematological variables analyzed would have placed the contributions of this study at a much higher level of impact. At this time, the authors could only report what occurs in each age group, without being able to evaluate any type of trend or behavior throughout the age range analyzed. In this sense, this reviewer only establishes the comment as a reflection, since it is considered that the contribution to the knowledge of the area of interest of this study is sufficiently justified with the broad cross-sectional sample used.

Fourth aspect to review

The discussion is well structured as long as the information between boys and girls is revised. However, the sentence indicated in lines 247-249 should be corrected: "Children with obesity have increased body fat mass, blood pressure, glucose, LDL-C, and blood cell counts, including WBC, RBC, and platelets". Regarding the variable "body fat mass", this statement is not possible based on the results of the study, since the authors have not analyzed this body composition variable. As noted, BMI is not a valid indicator to differentiate qualitative aspects of body mass composition.

Author Response

The study entitled “Positive association between body mass index and hematologic parameters, including RBC, WBC and platelet count, in Korean children and adolescent”, aims to investigate the associations between hematologic parameters and IMC in 7997 children and adolescents (4259 boys and 3738 girls) aged between 10 and 18 years old.

General Comments

This is a study that has its strong points in the large sample used and in the possibility of having been able to extract a wide variety of hematological variables. In this sense, the authors of the study are to be congratulated. On the other hand, although both the design and the methodology are also robust, there are a number of aspects that, in my opinion, would improve the information reported and are subject to major revision.

 à Thank you for your positive comment

First aspect to review

The object of study revolves around the relationship of a series of variables with BMI, an index that, although standardized and commonly used, has multiple limitations given that it is based exclusively on quantitative aspects of body mass without being able to discern the qualitative aspects that determine it (FFM, FM, etc.). Given that the BMI is also limited when it comes to establishing comparisons between different age groups, it is necessary that, in the introduction section, the authors include a paragraph on the advantages, but especially the limitations of the BMI: Nuttall FQ. Body Mass Index: Obesity, BMI, and Health: A Critical Review. Nutr Today. 2015 May;50(3):117-128.

 à Thank you for your comment. We have added the limitations of BMI to the revised text:

‘Third, because data regarding body composition and pubertal status were not available, the associations of body fat mass with hematological parameters were not analyzed. In addition, the effects of puberty on hematological changes and body composition were not evaluated. Importantly, BMI cannot differentiate between body lean and fat mass and poorly represents the body fat percentage and location’

Second aspect to review.

The manuscript in general is not entirely clear in expressing, at the same level of analysis, the results recorded for boys and girls separately. This is because, although at certain points in the text and in the tables the results are separated by sex, it seems that the choice of when and when not to do so has been made erratically at the choice of the authors. An example is Table 1. It is not done (and it is required by this reviewer) to show the results (and the consequent statistical treatment) for both sexes, and also for the whole group. Example of the reconfiguration of Table 1:   

à Thank you for your comment. We have analyzed the clinical characteristics according to sex and BMI. In addition, the statistical differences between boys and girls are presented in Supplementary Table 1 (supplementary file):  

All participants

Boys

Girls

Statistical differences between boys and girls

NW

OW

OB

NW

OW

OB

NW

OW

OB

(n = 6,421)

(n = 782)

(n = 794)

P

(n = 3350)

(n = 443)

(n = 466)

(n = 3071)

(n = 339)

(n = 328)

Pa

Pb

Pc

Boys

3,350

443

466

< 0.001

Age (y)

14.33 ± 2.51

14.24 ± 2.51

14.78 ± 2.54

< 0.001

14.35 ± 2.51

14.08 ± 2.49

14.56 ± 2.55

14.31 ± 2.51

14.44 ± 2.52

15.09 ± 2.50

0.498

0.042

0.004

Height SDS

0.17 ± 1.03

0.47 ± 1.04

0.53 ± 1.11

< 0.001

0.18 ± 1.04

0.55 ± 0.98

0.62 ± 1.07

0.16 ± 1.02

0.37 ± 1.11

0.40 ± 1.17

0.431

0.015

0.007

Weight SDS

–0.31 ± 0.89

1.29 ± 0.49

2.20 ± 0.74

< 0.001

–0.33 ± 0.90

1.31 ± 0.48

2.23 ± 0.74

–0.28 ± 0.87

1.26 ± 0.49

2.16 ± 0.73

0.043

0.461

0.160

BMI SDS (kg/m2)

–0.59 ± 0.88

0.86 ± 0.45

1.64 ± 0.75

< 0.001

–0.65 ± 0.88

0.87 ± 0.38

1.62 ± 0.65

–0.52 ± 0.86

0.85 ± 0.53

1.68 ± 0.88

< 0.001

0.962

0.278

WC SDS

–0.49 ± 0.85

1.32 ± 0.18

2.40 ± 0.67

< 0.001

–0.53 ± 0.87

1.32 ± 0.18

2.41 ± 0.65

–0.45 ± 0.83

1.32 ± 0.18

2.39 ± 0.68

< 0.001

< 0.001

0.673

SBP (mmHg)

105.65 ± 9.77

109.94 ± 10.28

113.68 ± 10.95

< 0.001

107.53 ± 10.09

112.69 ± 10.29

116.15 ± 10.73

103.60 ± 8.96

106.35 ± 9.10

110.18 ± 10.29

< 0.001

0.128

< 0.001

DBP (mmHg)

65.59 ± 8.91

66.88 ± 8.88

69.07 ± 9.19

< 0.001

65.82 ± 9.48

67.29 ± 9.75

69.45 ± 9.58

65.33 ± 8.25

66.34 ± 7.58

68.53 ± 8.59

0.028

0.527

0.157

WBC (× 103/mm3)

6.16 ± 1.49

6.65 ± 1.58

7.16 ± 1.65

< 0.001

6.13 ± 1.45

6.68 ± 1.53

7.13 ± 1.60

6.18 ± 1.53

6.61 ± 1.64

7.22 ± 1.71

0.163

< 0.001

0.432

RBC (× 106/mm3)

4.81 ± 0.39

4.89 ± 0.40

4.93 ± 0.41

< 0.001

5.02 ± 0.34

5.10 ± 0.34

5.13 ± 0.33

4.59 ± 0.31

4.62 ± 0.30

4.63 ± 0.31

< 0.001

< 0.001

< 0.001

Hemoglobin (g/dL)

13.97 ± 1.23

14.07 ± 1.22

14.19 ± 1.30

< 0.001

14.60 ± 1.11

14.64 ± 1.15

14.79 ± 1.20

13.28 ± 0.96

13.33 ± 0.85

13.34 ± 0.92

< 0.001

< 0.001

< 0.001

Hematocrit (%)

41.86 ± 3.40

42.28 ± 3.42

42.65 ± 3.53

< 0.001

43.43 ± 3.30

43.73 ± 3.38

44.18 ± 3.34

40.15 ± 2.60

40.38 ± 2.42

40.49 ± 2.50

< 0.001

0.431

< 0.001

Platelet (× 103/mm3)

282.96 ± 58.58

296.81 ± 59.92

309.03 ± 63.53

< 0.001

278.59 ± 58.48

295.33 ± 61.62

302.91 ± 61.66

287.73 ± 58.32

298.74 ± 57.65

317.73 ± 65.21

< 0.001

0.133

0.001

Glucose (mg/dL)

90.01 ± 7.39

91.83 ± 11.05

92.45 ± 12.16

< 0.001

90.69 ± 6.89

92.89 ± 13.37

92.53 ± 6.71

89.26 ± 7.84

90.44 ± 6.69

92.34 ± 17.16

< 0.001

0.001

0.851

T-C (mg/dL)

158.25 ± 26.27

163.00 ± 28.46

169.35 ± 29.50

< 0.001

153.81 ± 25.90

161.35 ± 29.78

168.12 ± 30.10

163.10 ± 25.81

165.17 ± 26.52

171.09 ± 28.57

< 0.001

0.059

0.163

HDL-C (mg/dL)

52.16 ± 9.94

47.68 ± 8.63

44.99 ± 8.37

< 0.001

51.13 ± 9.93

46.10 ± 8.33

44.27 ± 7.98

53.27 ± 9.82

49.75 ± 8.59

46.01 ± 8.80

< 0.001

< 0.001

0.004

TG (mg/dL)

79.29 ± 41.79

99.34 ± 54.29

112.75 ± 58.76

< 0.001

76.49 ± 42.07

103.29 ± 58.81

110.92 ± 58.64

82.33 ± 41.26

94.17 ± 47.33

115.34 ± 58.91

< 0.001

0.017

0.297

LDL-C (mg/dL)

90.24 ± 22.42

95.46 ± 25.07

101.81 ± 25.63

< 0.001

87.38 ± 22.08

94.59 ± 26.15

101.67 ± 26.28

93.36 ± 22.38

96.58 ± 23.57

102.02 ± 24.72

< 0.001

0.264

0.849

Alcohol use

1564 (24.36%)

188 (24.04%)

237 (29.85%)

0.003

888 (26.51%)

110 (24.83%)

142 (30.47%)

676 (22.01%)

78 (23.01%)

95 (28.96%)

< 0.001

0.613

0.705

Smoker

718 (11.18%)

88 (11.25%)

106 (13.35%)

0.191

524 (15.64%)

61 (13.77%)

77 (16.52%)

194 (6.32%)

27 (7.96%)

29 (8.84%)

< 0.001

0.015

0.002

Household income ≤ 1st quartile

687 (10.70%)

82 (10.49%)

95 (11.96%)

0.531

362 (10.81%)

39 (8.80%)

50 (10.73%)

325 (10.58%)

43 (12.68%)

45 (13.72%)

0.804

0.102

0.243

Rural residence

985 (15.34%)

113 (14.45%)

125 (15.74%)

0.757

518 (15.46%)

60 (13.54%)

69 (14.81%)

467 (15.21%)

53 (15.63%)

56 (17.07%)

0.803

0.471

0.445

Physical activity

2368 (36.88%)

279 (35.68%)

316 (39.80%)

0.194

1351 (40.33%)

171 (38.60%)

183 (39.27%)

1017 (33.12%)

108 (31.86%)

133 (40.55%)

< 0.001

0.061

0.773

Hypertension

1 (0.02%)

1 (0.13%)

1 (0.13%)

0.124

1 (0.03%)

1 (0.23%)

 0 (0%)

0 (0%)

0 (0%)

1 (0.3%)

> 0.999

> 0.999

0.860

T2DM

0 (0%)

0 (0%)

0 (0%)

> 0.999

0 (0%)

0 (0%)

0 (0%)

0 (0%)

0 (0%)

0 (0%)

> 0.999

> 0.999

> 0.999

Dyslipidemia

0 (0%)

0 (0%)

0 (0%)

> 0.999

0 (0%)

0 (0%)

0 (0%)

0 (0%)

0 (0%)

0 (0%)

> 0.999

> 0.999

> 0.999

 In accordance with this, the entire manuscript should be revised to balance the expression of results, discussion and conclusions at the same level between both sexes. This is critical in this study because, as the authors themselves recognize, one of the aspects that most limit the consistency of their results and consequent interpretations, is the fact that no biological age variable has been recorded, a circumstance that is aggravated by the age range analyzed (10 - 18 years...), where the internal changes between boys and girls during puberty are significant and limit, to the point of being considered an error, to report them jointly....

 à Thank you for your comment. We have added the following text to the revised manuscript (Discussion):

 ‘ The mechanisms underlying the sex differences in blood cell composition and their effects on the risk of obesity-related complications are not well understood. It is possible that these findings result from differences in body fat composition between males and females. Nuttal reported that BMI may not be an accurate marker of obesity because men tend to accumulate fat in the abdominal area, whereas women tend to accumulate it in the peripelvic area and thighs. Prior to puberty, boys and girls have similar patterns of body fat deposition; however, during and after puberty, girls tend to accumulate a large quantity of fat, whereas boys accumulate a large quantity of lean mass (bone and muscle) but not fat mass. These changes lead to an increased BMI in both sexes. The hematological parameters of boys and girls are different between those tested before and after puberty. In the present study, information related to the body fat mass and pubertal status was not available. Therefore, further studies are needed to determine the sex differences in hematological parameters before and after puberty.’

Third aspect to review (or reflect on)

Although the authors recognize the limitations of the cross-sectional design, this could have been overcome by including a longitudinal follow-up subsample. That is, under a temporal inclusion criterion (e.g., including subjects analyzed for "X" years in a row), add their analysis. Undoubtedly, determining the "cause-effect" relationship of BMI behavior with the hematological variables analyzed would have placed the contributions of this study at a much higher level of impact. At this time, the authors could only report what occurs in each age group, without being able to evaluate any type of trend or behavior throughout the age range analyzed. In this sense, this reviewer only establishes the comment as a reflection, since it is considered that the contribution to the knowledge of the area of interest of this study is sufficiently justified with the broad cross-sectional sample used.

à Thank you for your useful comments. We agree with your comments that a longitudinal study would prove the cause-and-effect relationship between obesity and hematological variables. However, although the KNHANES included a representative population group from Korea, a longitudinal study could not be performed because the study participants changed every year. Using the current data, it is possible to conduct familial research using codes that identify families.

Fourth aspect to review

The discussion is well structured as long as the information between boys and girls is revised. However, the sentence indicated in lines 247-249 should be corrected: "Children with obesity have increased body fat mass, blood pressure, glucose, LDL-C, and blood cell counts, including WBC, RBC, and platelets". Regarding the variable "body fat mass", this statement is not possible based on the results of the study, since the authors have not analyzed this body composition variable. As noted, BMI is not a valid indicator to differentiate qualitative aspects of body mass composition.

  à Thank you for your comments. We agree with your recommendation and have modified the sentence as follows:

 ‘OB children had higher blood pressure, glucose, LDL-C, WBC, RBC, and platelet levels.’

Reviewer 3 Report

I've got some questions or suggestions refers to the manuscript:

1) Abstract should be a single text, without sections

2) What other measurements were taken in addition to the blood test which the authors write about in line 69?

3) What were the inclusion criteria (only the exclusion criteria are described)

4) Please provide the exact methodology of anthropometric measurements, in what position, with or without shoes, after rest or not, or on fasting status? These are very important details

5) Adapt the references to the journal's requirements and write citations in square brackets, not round brackets.

6) Very insightful and detailed discussion, correctly written.

Author Response

I've got some questions or suggestions refers to the manuscript:

1) Abstract should be a single text, without sections

à Thank you for your comment. We agree with your recommendation and have modified the abstract.

2) What other measurements were taken in addition to the blood test which the authors write about in line 69?

à KNHANES is conducted every 3 years, and the present study included data from the fourth (2007–2009), fifth (2010–2012), and sixth (2013–2015) surveys, as well as the first and second years of the seventh survey (2017–2018). Height, weight, blood pressure, waist circumference, and pulse were recorded annually. Different blood tests were conducted in each survey. The levels of glucose, HbA1c, total cholesterol, triglyceride, HDL cholesterol, LDL cholesterol (measured directly), HBsAg, AST, ALT, HCV titer, WBCs, RBCs, Hb, Hct, platelets, BUN, and creatinine were measured annually. From 2017 to 2018, the levels of hsCRP, uric acid, folate, vitamin A, and vitamin E were measured in the seventh survey. In addition, the levels of urinary pH, nitrite, specific gravity, protein, glucose, ketone, cotinine, sodium, potassium, and heavy metals (including cadmium and mercury) were evaluated every year. An oral examination and pulmonary function test were also performed.

3) What were the inclusion criteria (only the exclusion criteria are described)

à Boys and girls aged 10–18 years were included in the analysis. We attempted to include a large number of representative Korean children and adolescents.

4) Please provide the exact methodology of anthropometric measurements, in what position, with or without shoes, after rest or not, or on fasting status? These are very important details

à Thank you for your comments. We have modified the text as follows:

‘Anthropometric assessments, including height, weight, waist circumference (WC), and systolic and diastolic BP (SBP and DBP, respectively), were performed by an expert. Height was measured to the nearest 0.1 cm using the standard method on a flat floor without shoes or bulky clothing. The participants stood with their back toward the measuring rod, face forward, feet placed together, knees straight, heels touching the heel plate or wall, and shoulders, buttocks, and head inside the stadiometer (Seca, Hamburg, Germany). Body weight was measured to the nearest 0.1 kg while wearing light clothing using an electronic scale (G-tech, Seoul, Korea). WC was measured to the nearest 0.1 cm at the level between the lower rib margin and iliac crest using a calibrated measuring tape (Seca). SBP and DBP were measured three times at an interval of 2 min to the nearest 1 mmHg from the right upper arm using a calibrated sphygmomanometer. The mean values of the final two SBP and DBP measurement were used for analysis.’

5) Adapt the references to the journal's requirements and write citations in square brackets, not round brackets.

à Thank you for your comments. We agree with your recommendation and have modified the reference style.

6) Very insightful and detailed discussion, correctly written.

à Thank you for your positive comments

Round 2

Reviewer 1 Report

I accept the authors' explanation.

Author Response

Thank you so much

Reviewer 2 Report

Dear authors,

Thanks for the effort to attend the contributions made by my review. Congratulations for the work done.

Author Response

Thank you so much

Reviewer 3 Report

The authors attached the file with corrections without using the journal template.

Please add answers to my comments 2 and 3 to the text of the manuscript

Author Response

The authors attached the file with corrections without using the journal template.

à We have modified the reference style (ACS style)

Please add answers to my comments 2 and 3 to the text of the manuscript

à Thank you for your comments. We have added the answer to the Materials and Methods

‘KNHANES enrolled 97,622 individuals between 2007 and 2017. Of these, our preliminary analyses included 10,734 (5,670 boys, 5,064 girls) aged 10–18 years (Figure 1). We attempted to include a large number of representative Korean children and adolescents. After obtaining informed consent, the participants and their parents were interviewed at their homes, and the participants underwent several examinations, including blood sampling, an oral examination, and pulmonary function tests. Different blood tests were conducted in each survey. The levels of glucose, HbA1c, total cholesterol, triglyceride, HDL cholesterol, LDL cholesterol (measured directly), HBs Ag, AST, ALT, HCV titer, BUN, and creatinine were measured annually. In the seventh survey from 2017 to 2018, the levels of hsCRP, uric acid, folate, vitamin A, and vitamin E were measured. In addition, the levels of urinary pH, nitrite, specific gravity, protein, glucose, ketone, cotinine, sodium, potassium, and heavy metals (including cadmium and mercury) were evaluated every year. The hematological parameters recorded were the WBC, RBC, Hb, Hct, and platelet count. Participants with incomplete physical examination records, including incomplete anthropometric measurements and laboratory test results (such as lipid profiles), or triglyceride levels ≥ 400 mg/dL were excluded (n = 17). The KNHANES database is publicly available (http://knhabes.cdc.go.kr). The 2007–2018 KNHANES study protocols were approved by the Institutional Review Board of the Korean Centers for Disease Control and Prevention. Informed consent was provided by the KNHANES participants.

Anthropometric assessments, including height, weight, waist circumference (WC), and systolic and diastolic BP (SBP and DBP, respectively), were performed by an expert and recorded annually.~’